# Antineoplastic Activity of *Rhus trilobata* Nutt. (*Anacardiaceae*) against Ovarian Cancer and Identification of Active Metabolites in This Pathology

**DOI:** 10.3390/plants10102074

**Published:** 2021-09-30

**Authors:** Luis Varela-Rodríguez, Blanca Sánchez-Ramírez, Erika Saenz-Pardo-Reyes, José Juan Ordaz-Ortiz, Rodrigo Daniel Castellanos-Mijangos, Verónica Ivonne Hernández-Ramírez, Carlos Martín Cerda-García-Rojas, Carmen González-Horta, Patricia Talamás-Rohana

**Affiliations:** 1Facultad de Enfermería y Nutriología, Universidad Autónoma de Chihuahua, Chihuahua CP 31125, CHIH, Mexico; lvrodriguez@uach.mx (L.V.-R.); erika.saenzpardo@gmail.com (E.S.-P.-R.); 2Facultad de Ciencias Químicas, Universidad Autónoma de Chihuahua, Chihuahua CP 31125, CHIH, Mexico; carmengonzalez@uach.mx; 3Laboratorio de Metabolómica y Espectrometría de Masas, Unidad de Genómica Avanzada—CINVESTAV, Irapuato CP 36824, GTO, Mexico; jose.ordaz.ortiz@cinvestav.mx; 4Servicio de Imagenología Diagnóstica, ISSEMyM Arturo Montiel Rojas, Metepec CP 52170, MEX, Mexico; rodan_casmij@hotmail.com; 5Departamento de Infectómica y Patogénesis Molecular, Centro de Investigación y de Estudios Avanzados del Instituto Politécnico Nacional (CINVESTAV-IPN), Ciudad de México CP 07360, CDMX, Mexico; arturomvi@hotmail.com; 6Departamento de Química, Centro de Investigación y de Estudios Avanzados del Instituto Politécnico Nacional (CINVESTAV-IPN), Ciudad de México CP 07360, CDMX, Mexico; ccerda@cinvestav.mx

**Keywords:** antineoplastic activity, gallic acid, myricetin, ovarian cancer, *Rhus trilobata*

## Abstract

*Rhus trilobata* (RHTR) is a medicinal plant with cytotoxic activity in different cancer cell lines. However, the active compounds in this plant against ovarian cancer are unknown. In this study, we aimed to evaluate the antineoplastic activity of RHTR and identify its active metabolites against ovarian cancer. The aqueous extract (AE) and an active fraction (AF02) purified on C_18_-cartridges/ethyl acetate decreased the viability of SKOV-3 cells at 50 and 38 μg/mL, respectively, compared with CHO-K1 (>50 μg/mL) in MTT assays and generated changes in the cell morphology with apoptosis induction in Hemacolor^®^ and TUNEL assays (*p* ≤ 0.05, ANOVA). The metabolite profile of AF02 showed a higher abundance of flavonoid and lipid compounds compared with AE by UPLC-MS^E^. Gallic acid and myricetin were the most active compounds in RHTR against SKOV-3 cells at 50 and 166 μg/mL, respectively (*p* ≤ 0.05, ANOVA). Antineoplastic studies in *Nu/Nu* female mice with subcutaneous SKOV-3 cells xenotransplant revealed that 200 mg/kg/i.p. of AE and AF02 inhibited ovarian tumor lesions from 37.6% to 49% after 28 days (*p* ≤ 0.05, ANOVA). In conclusion, RHTR has antineoplastic activity against ovarian cancer through a cytostatic effect related to gallic acid and myricetin. Therefore, RHTR could be a complementary treatment for this pathology.

## 1. Introduction

Ovarian cancer is the sixth most frequent tumor in women and the fourth cause of death in Mexico due to gynecological tumors [1]. Ovarian tumors can be primary or metastatic and are classified based on their origin as epithelial, germinal, or stromal of the sexual cord tumors [2]. Epithelial tumors are the most common type of ovarian neoplasm; among them, the serous subtype usually appears more frequently [2]. Surgical resection is the principal treatment for this disease, followed by antitumoral chemotherapy with cytotoxic or cytostatic drugs [1,3]. However, the surgent of resistance in neoplastic tissue limits ovarian cancer chemotherapy’s success [3], making it necessary to search for alternative treatments or new therapeutic agents for this disease. Plants used for alternative medicine represent an option in the search for active compounds for cancer treatment. Recent work with *Rhus trilobata* Nutt. (RHTR) demonstrated the presence of antineoplastic agents that could be new therapeutic candidates for the treatment of ovarian cancer that afflicts the female population in Mexico and around the world [4].

In Mexico, RHTR (*Anacardiaceae* family) is used for the treatment of gastrointestinal diseases and cancer. RHTR is known by the common name of *aciditos* or *agritos* due to the characteristic flavor of its fruit [4]. Previous studies by Abbott et al. (1966) demonstrated the antineoplastic activity of RHTR in Syrian hamsters xenotransplanted with duodenum adenocarcinoma. Animals were treated with extracts of leaves (100 mg/kg/i.p.) or fruits (400 mg/kg/i.p.) for seven days; tumoral lesions decreased 33% with both extracts in comparison with control conditions [5]. Subsequently, Pettit et al. (1974) isolated gallic acid (Ga) from RHTR leaves with column chromatography (Sephadex LH-20) and determined its biological activity; Ga decreased KB cell viability (IC_50_: 3.1 μg/mL). The authors concluded that the medicinal properties of RHTR correspond to this compound [6]. Studies conducted by our research group revealed that the aqueous extract (AE) of RHTR stems contains quinic acid, myricetin (Myr), Ga, 1,2,3,4,6-pentakis-*O*-gallioyl-*β*-D-glucose (*β*-PGG), quercetin, obtusaquinol, fisetin, margaric acid, and amentoflavone by UPLC-MS^E^, which are compounds with biological activity already demonstrated by other studies [4]. Additionally, AE-RHTR presented a selective activity against CACO-2 cells (IC_50_: 5 μg/mL), and low toxicity (LD_50_: 1141.5 mg/kg) [4]. Thus, the biological activity observed in RHTR may be related to compounds such as phenolic acids or flavonoids and the synergic effects between both molecule types. 

Therefore, we aimed to evaluate the antineoplastic activity of RHTR in mice xenotransplanted with a human ovarian cancer cell line and to identify the active metabolites against this pathology.

## 2. Results and Discussion

### 2.1. Biological Activity of AE-RHTR and Fractions in Cell Lines

The AE and fractions from RHTR presented biological activity in SKOV-3 cells at 50 μg/mL (Figure 1A). In comparison, the concentration required to affect the viability of CHO-K1 cells was greater than 50 μg/mL (Figure 1B), demonstrating a selective effect. AE and AF02 were most active against SKOV-3 cells at 50 and 38 μg/mL, respectively, compared with 1× PBS (vehicle group) (*p* < 0.05, Dunnett). However, AE and AF02 were found to only have a 24 h limited inhibitory effect on SKOV-3 cell proliferation (Figure 1C). Additionally, SKOV-3 cells treated with AE and AF02 had a similar morphology to the vehicle group but with a considerable increase in cytoplasmic vesicles and an absence of cellular mitosis, which suggest a quiescent effect in both treatments (Figure 1D). The apoptosis assays mainly demonstrated increased caspase-3/7 activity and nuclear DNA-fragmentation in cells treated with AE (8.47% ± 0.9%), AF02 (14.01% ± 3.7%), and paclitaxel (46.42% ± 5.0%) at 24 h in comparison with the vehicle group (2.47% ± 0.4%) (*p* < 0.05, ANOVA; Figure 1D,E). Necrotic events were present during treatments with both samples at 72 h (Figure 1E), possibly related to stages of late apoptosis and not with true necrosis, which can promote inflammatory processes; however, additional studies are necessary to corroborate this finding. These results resemble those obtained with CACO-2 cells, where cell cycle arrest in G_1_ and the appearance of a G_1_ subpopulation related to apoptosis were observed. In contrast, in BEAS-2B cells, increasing the concentration up to 800 μg/mL was necessary to observe a similar effect [4]. Therefore, AE and AF02 were selected to evaluate their antineoplastic activity.

### 2.2. Antineoplastic Activity of RHTR in Mice Xenotransplanted with SKOV-3 Cells

After 24 h of treatment, the rodents showed no behavioral changes. Additionally, the analyses of their bodyweight showed that the group treated with AE experienced a 9.5% decrease, but a 3.7% increase in the group with AF02. However, these changes were not significant compared to the control group treated with 1× PBS vehicle (*p* > 0.05, Dunnett; Figure 2A). At the end of treatments, mice were euthanized for tumor lesions recovery and to perform a macroscopic analysis of the developed lesions (Figure 2C). Similar characteristics in all tumors were present in the treated groups with an ovoid or smooth-surfaced morphology and the presence of vascularity (Figure 2C). However, minor differences in the color of tumors were found: pink for AE and whitish for AF02; groups treated with carboplatin and vehicle presented a yellow tumor with a nodular surface (Figure 2C). Analysis of tumoral weight showed that greater masses were found in the vehicle group, followed by AE, AF02, and carboplatin (*p* > 0.05, Tukey; Table 1). These results correlate with the tumor volume of lesions present in rodents along with the treatment time (Figure 2B). The inhibition percentage in tumoral growth with AE, AF02, and carboplatin was 37.6%, 49%, and 74.5%, respectively, compared with the control group for 14 and 28 days of treatment (*p* ≤ 0.05, Dunnett; Figure 2B,C). These findings demonstrate changes in the disease evolution directly related to the treatments used in the study. Similarly, these results agree with those obtained in tumoral length (Table 1), which demonstrated a cytostatic effect of the treatments.

#### 2.2.1. Imaging and Histopathological Studies of Tumor Lesions

Images by NMR showed ovoid tumors located in the subcutaneous tissue, with attenuation units related to soft tissues (≈50 HU; Table 1). Treatments with AE and AF02 induced regular edges in tumoral lesions, whereas groups treated with carboplatin and vehicle were mainly lobulated edges (Figure 3A). Additionally, the necrosis percentage was higher in mice treated with vehicle (35.16%), whereas those treated with AE, AF02, and carboplatin were 30.68%, 20.54%, and 5.19%, respectively (Figure 3A). Images by USG corroborated the presence of subcutaneous ovoid tumors, with partially defined edges and heterogeneous content (hyperechoic areas with diffuse distribution concerning fibrosis) in the different groups (Figure 3B). Vascularity presence was observed in all tumors (seen as red and blue color); however, the highest vascularity was found in mice treated with vehicle compared with AE, whereas the lower vascularity was observed in those animals treated with AF02 or carboplatin (Figure 3B). The histological analysis of cystic lesions located in the reticular dermis of rodents revealed a stroma/parenchyma of mixed composition and without residual organ (Figure 3C). The abundant presence of loose fibrous tissue was observed in all tumors with a predominance of comedo-type necrosis for the vehicle (25%), decreasing by AE and AF02 treatment (10%), or basaloid-type histological pattern after carboplatin (Figure 3C, H&E). Additionally, the adjacent stroma in tumors presented retraction artifacts (cracks) and signs of acute/moderate chronic inflammation by variable cellular infiltrate (Figure 3C, H&E). Other characteristics observed in all tumors were the proliferation of basaloid cells (basement membrane cells-like) organized in a solid pattern that grew to push the stroma, as well as palisade cells on the tumor periphery with a radial orientation at superior parallel axes (Figure 3C, H&E and TOB). Similarly, most cells presented mixed nuclei (small and hyperchromatic) with vesicular chromatin and a poorly defined wide eosinophilic cytoplasm (Figure 3C, H&E). RHTR treatments induced atypical mitosis and apoptosis in comparison with the control group (Figure 3C, TOB and TEM). All these characteristics observed in tumors correspond to poorly differentiated carcinomas (possibly of the serous papillary type) [7]. The first-choice treatment for ovarian cancer (with anaplasia IV degree) is paclitaxel and carboplatin since both compounds exert their effects through different mechanisms of action [1]. Carboplatin can generate DNA adducts to prevent cell proliferation [8], and paclitaxel can bind to *β*-tubulin to stabilize the microtubules and block the mitosis, which triggers cell death by apoptosis [9]. Therefore, the similar effect that induced the RHTR compounds present in AE and AF02 to drugs used in chemotherapy for ovarian cancer (particularly with carboplatin) makes them suitable candidates for more comprehensive structural studies and sheds light on their potential application against this disease.

#### 2.2.2. Morphometric and Paraclinical Studies of Mice Treated with RHTR

Anatomical lesions (Figure 3D) or morphometric changes (Appendix A) in the liver and kidneys were absent in mice treated with RHTR. Additionally, histopathological lesions in the organs were not found by comparison with the control group (1× PBS; Figure 3D). Finally, paraclinical studies revealed slight leukopenia in mice treated with AE and AF02 at 28 days of administration concerning reference values for mice (*p* ≤ 0.05, ANOVA; Table 2). The biochemical analysis demonstrated a decrease in albumin for mice treated with AE and AF02, whereas urea decreased with AE treatment at the end of the study (*p* ≤ 0.05, ANOVA; Table 2). The results suggest slight hematopoiesis suppression in the bone marrow with AE and AF02 treatments, possibly by the quickly replicative phenotype of blood cells [10]. Additionally, paraclinical studies did not reveal hepatic dysfunction or renal failure, so the use of RHTR as an alternative treatment for ovarian cancer could be considered.

### 2.3. Purified Metabolites in RHTR

Methyl gallate (methyl 3,4,5-trihydroxybenzoate, C_8_H_8_O_5_) was purified from RHTR and identified by nuclear magnetic resonance (NMR). The signal obtained was ^1^H-NMR (300 MHz, CD_3_OD) δ: 7.07 (H-2, H-6), δ: 4.9 (H-3, H-4, H-5), δ: 3.8 (H-8) (Appendix A). ^13^C-NMR (300 MHz, CD_3_OD) δ: 122.3 (C-1), δ: 110.9 (C-2, C-6), δ: 147.3 (C-3, C-5), δ: 140.6 (C-) 4), δ: 169.8 (C-7), δ: 53.1 (C-8) (Appendix A). The identification pattern obtained is similar those reported by other authors [13,14]. Methyl gallate is present in several species of the *Rhus* genus [15] and exhibited antioxidant, antimicrobial, and antineoplastic activity in various investigations [16,17,18]; therefore, the biological activity of methyl gallate was evaluated in SKOV-3 cells.

### 2.4. Phytochemical Composition and in Silico Analysis of AF02-RHTR

The metabolite profile of AF02 showed a higher abundance of flavonoid and lipid compounds (Table 3). Additionally, a comparative study between the metabolite profile of AE and AF02 showed that epigallocatechin 3-cinnamate (3), 3,5-digalloylepicatechin (4), Myr (5), myricitrin (6), quercetin (9), hibiscoquinone A (11), quercitrin (12), myricetin 3-(4’’-galloylrhamnoside) (13), obtusaquinol (14), epifisetinidol-(4*β*→8)-catechin (15), 12*S*-hydroxy-16-heptadecynoic acid (18), (-)-pinellic acid (19), 11,14-eicosadienoic acid (20), amentoflavone (21), 3-(1,1-dimethylallyl)-8-(3,3-dimethylallyl)xanthyletin (22), lignocerate (23), and 2*R*-hydroxy-9*Z*,12*Z*-octadecadienoic acid (24) had a high relative abundance (double) in AF02 (Table 3, Figure 4A). These compounds were identified by their fragmentation patterns and the matching of their retention times with analytical standards (Appendix A). A comprehensive analysis of the fractions revealed that compound 5 (RT: 8.27 min) was present in all fractions and had higher relative abundance in AF02 (Figure 4A). Myr is a compound that was reported to have cytotoxic activity against different cancer cell lines [19]. Therefore, Myr could be related to the antineoplastic activity of RHTR on ovarian cancer studied in this project. To delimit the scope of this study, the experimental design focused on an in silico analysis and bibliographic search of the medicinal properties of the major metabolites in AF02-RHTR to determine its possible activity against ovarian cancer (Table 3). However, the choice of this design most likely generated a bias in which possibly, unintentionally, some active compound in low concentration or the synergistic effect that can be generated by the combined presence of several compounds were omitted. This opens the possibility for future studies aimed at elucidating these two possibilities, either by our own group or by members of the community interested in studying natural compounds with potential anticancer bioactivity. Compounds that presented a higher score in the drug-likeness model were: 4 (DLMS: 1.52), 6 (DLMS: 0.78), 9 (DLMS: 0.93), 12 (DLMS: 1.04), 13 (DLMS: 1.12), 15 (DLMS: 0.61), and 21 (DLMS: 0.51) (Table 3). However, Myr presented a –0.04 value, possibly due to the low specificity of its pharmacological effect. These compounds are classified as flavonoids and were identified in other species of the *Rhus* genus, which mainly show antioxidant, anti-inflammatory, and cytotoxic activities (Table 3) [15]. Consequently, compounds 5 (RT: 8.27 min), 6 (RT: 8.47 min), 9 (RT: 8.60 min), 12 (RT: 9.45 min), and 16 (RT: 11.92 min) were selected to further evaluate their cytotoxic activity against SKOV-3 cells.

### 2.5. Biological Activity of Active Metabolites from RHTR in Cell Lines

The biological activity of purified compounds in RHTR such as Ga [6] and methyl gallate (2) was evaluated. Additionally, compounds selected from the in silico analysis, such as 5, 6, 9, 12, and 16, were evaluated in cancer and normal epithelial cells. The results showed that the compounds had biological activity in SKOV-3 (Figure 4B) and OVCAR-3 (Appendix A) cell lines at concentrations lower than 200 μg/mL (Table 4). However, Ga (50/43 μg/mL) and Myr (166/94 μg/mL) had the highest activity compared with the vehicle group (0.5% DMSO) (*p* ≤ 0.05, Dunnett). The biological activity of both compounds was corroborated in CACO-2 cells (Appendix A) at 25 and 62 μg/mL, respectively (*p* ≤ 0.05, Tukey; Table 4). Similarly, both compounds presented cytotoxic activity in CHO-K1 and BEAS-2B cells (Appendix A) at ~150 μg/mL, possibly through a genotoxic effect related to reactive oxygen species (ROS) production [20,21]. The highest activity was observed with Myr at 33 μg/mL in CHO-K1 and Ga at 25 μg/mL in BEAS-2B; the lowest activity was found with quercitrin and methyl gallate at 200 μg/mL in both cell lines (*p* ≤ 0.05, Dunnett) (Table 4). These results demonstrated that the biological activity of the compounds considerably depends on the cellular phenotype. Cell morphology analysis in SKOV-3 revealed that the compounds induced cellular changes such as individualization, rounding, microfilaments loss, cytoplasm condensation, nuclear fragmentation, and cell monolayer breakdown after 24 h of treatment, and these effects were similar to those for the paclitaxel group (Figure 4C). Cells treated with vehicle (0.5% DMSO) presented lamellipodia, filopodia, stress fibers, adhesion dots, mitosis, confluence, and cell monolayer formation (Figure 4C). 

The results showed that the principal metabolites in RHTR against ovarian cancer cells could be Ga and Myr. These compounds are active in multiple cancer cell lines through ROS generation (amongst other processes) that triggers cell death by apoptosis [19,22]. Complementary studies by our research group in mouse models for ovarian cancer demonstrated that Ga and Myr at 50 mg/kg administered by the peritumoral route inhibited tumoral lesions (50% of progression), decreased vascularity, and induced apoptosis, with few toxicological effects, possibly by a mechanism related to carbonic anhydrase-IX or PI3K [23]. However, there is the possibility that other metabolites in RHTR (majority or present) could have specific activity against different cancer types because cancer cells can develop a sensitivity or selective resistance to certain drugs classes related to the phenotypic and functional heterogeneity, as well as molecular features present in each neoplasm disease [24]. Therefore, these findings could be a perspective to address in future studies. Similarly, additional studies are required to demonstrate if the antineoplastic activity observed in RHTR results from a synergistic mechanism among other abundant compounds in the plant such as *β*-PGG (Table 3) [25,26]. Recent studies have demonstrated that *β*-PGG can induce cell cycle arrest in breast cancer (MCF-7, 50 μM) in the G_1_ phase by inhibiting kinase activity in the D/CDK4 and CDK2 complex, decreasing the phosphorylation in retinoblastoma protein (pRB), and increasing the levels of p27Kip, p21Cip, and p53 [27,28]. Thus, *β*-PGG could be involved in the biological activity found in RHTR; however, more in-depth studies are required to confirm this claim. Finally, the results show that RHTR, Ga, and Myr could play a role in the alternative treatment of ovarian cancer.

## 3. Materials and Methods

### 3.1. Chemicals and Reagents

Compounds evaluated were Ga (G7384), methyl gallate (274194), Myr (M6760), myricitrin (91255), quercetin (Q4951), quercitrin (Q3001), and fisetin (F4043) from Sigma-Aldrich© (St. Louis, MO, USA) (HPLC-grade). Control drugs were paclitaxel (5 µg/mL in cells; T7402, Sigma^®^), vincristine (20 µg/mL in cells; V8879, Sigma^®^, St. Louis, MO, USA), and carboplatin (50 mg/kg/3 alternating days/week in mice; C2538, Sigma^®^, St. Louis, MO, USA); all drugs are chemotherapeutic agents used in treatment against ovarian cancer. Vehicle controls were 1× PBS (100 µL/day in animals) or 0.5% DMSO-1X PBS (*v/v* in cells; D2650, Sigma^®^, St. Louis, MO, USA). Additional use of equipment and reagents are indicated in the text.

### 3.2. Recollection of the Plant Material

RHTR Nutt. (Common name: skunkbush sumac; Family: Anacardiaceae; The Plant List: TRO-1300191) was collected in May 201, from Namiquipa municipality, Chih., Mexico (Location: Cerro Pelón, Colonia Independencia; 29°5’59’’ N, 107°32’33’’ W). Community owners kindly donated the RHTR samples. Toutcha Lebgue-Keleng (FZE-UACH) performed the taxonomic identification of collected specimens. RHTR was validated and incorporated into the herbarium from Escuela Nacional de Ciencias Biológicas (ENCB-IPN). Finally, RHTR stems were milled until 0.5 mm to be lyophilized and preserved by refrigeration [4].

### 3.3. Preparation of the Plant Extract and Fractionation

An aqueous extract (AE) of RHTR was elaborated by decoction (25 g of stems and 500 mL of boiling distilled water, 30 min), centrifugation (2500 rpm, 15 min, 4 °C), concentration (negative pressure, 40 °C, 5 rpm) (Rotavapor^®^ R-300, Büchi, Flawil, C.H.), and lyophilization (freezing: –40 °C, 2 h; sublimation: –15 °C, 3 h; desorption: 40 °C, 1 h; temperature ramp: 1 °C/min) (FreeZone Triad, Labconco^®^, Kansas City, MO, USA). Later, AE was fractionated with ENVI™-C_18_ cartridges (Supelclean™, Sigma^®^, St. Louis, MO, USA) in a vacuum manifold (Visiprep™, Sigma^®^, St. Louis, MO, USA). The solvents used were 1% acetic acid (aqueous fraction-01; AF01), ethyl acetate (aqueous fraction-02; AF02), and ethylic ether (aqueous fraction-03; AF03) (15 mL each, HPLC grade, J.T. Baker^®^, Edo. de México, M.X.). Finally, fractions were evaporated, weighed, resuspended in 2 mL 50% MeOH (MS grade, J.T. Baker^®^, Edo. de México, M.X.), and filtered with a 0.22 µm PTFE syringe filter (Corning^®^, Corning, NY, USA). Extract and fractions were prepared according to the method disclosed in the Mexican patent MX/E/2018/078316.

### 3.4. Purification and Identification of Metabolites

The compounds in AE-RHTR were purified by open-column chromatography (Econo-Column^®^, Bio-Rad^®^, Hercules, CA, USA), with 1 g silica gel (60 Å, 230-400 mesh, Merck^®^, Darmstadt, D.E.) and a mobile phase of CHCl_3_:MeOH (9:1, 8:2, 7:3, 1:1, 3:7 and 100% MeOH). With a UV-fraction collector (CF-2, Spectrum^®^, New Brunswick, NJ, USA), we obtained thirty-six fractions (20 mL/each), and through NMR spectrometry (Mercury, Varian^®^, Palo Alto, CA, USA), we identified the compounds. Monodimensional assays (^1^H-NMR and ^13^C-NMR) were performed with methanol-4d (CD_3_OD, Sigma^®^, St. Louis, Missouri, USA) at 300 MHz and 25 °C. Tetramethylsilane (TMS, Sigma^®^, St. Louis, MO, USA) was the internal reference. Finally, in NMR spectra demonstrated the chemical shifts (δ) as ppm from the TMS signal.

### 3.5. Cell Culture

The cell lines used were SKOV-3 (HTB-77™), OVCAR-3 (HTB-161™), CHO-K1 (CCL-61™), CACO-2 (HTB-37™), and BEAS-2B (CRL-9609™), acquired from ATCC^®^ (Virginia, USA). The cell monolayers were maintained according to the supplier’s specifications at 37 °C and 5% CO_2_.

### 3.6. Determination of Cell Viability by Formazan Salts

Cultures (2 × 10^4^ cells/well) were seeded with 200 μL of supplemented medium (Sigma^®^) in 96-well plates (Corning^®^, Corning, NY, USA) and incubated for 24 h. Cell treatments were 5-200 μg/mL of samples/controls for 24 h. At the end of the treatments, we added MTT (5 mg/mL, Sigma^®^) for 4 h of incubation, and optical density was measured at λ = 590 nm in a VariosKan^®^ Flash (Thermo-Scientific^®^, Waltham, MA, USA). The half-maximal inhibitory concentration (IC_50_) was determined [4].

### 3.7. Measurement of Proliferation, Cytotoxicity, and Apoptosis in Cells

Cells (2 × 10^4^/per well) were cultivated in 96-well black plates (Corning^®^, Corning, NY, USA) with 100 μL of supplemented medium (Sigma^®^, St. Louis, MO, USA) and incubated for 24 h. Treatments on adherent cells were conducted at 5-20 μg/mL of samples/controls for 24 h. Then, ApoTox-Glo™ Triplex Assay (Promega^®^, Madison, WI, USA) was used to determine events of viability (proliferation), cytotoxicity (necrosis), and apoptosis (caspase-3/7 activation). Additionally, apoptosis was confirmed by an ApoAlert™ DNA Fragmentation Assay (Clontech^®^, Kusatsu, Shiga, J.P.). Both assays were performed according to each manufacturer’s instructions. Finally, optical density was measured in a VariosKan^®^ Flash (Thermo-Scientific^®^, Waltham, MA, USA) [29].

### 3.8. Evaluation of Cell Morphology

Cells (3 × 10^4^/per well) were seeded in Lab-Tek™ chamber slides (Thermo-Scientific^®^, Waltham, MA, USA) with 400 µL of supplemented medium (Sigma^®^, St. Louis, MO, USA) and incubated for 24 h. The treatment duration with IC_50_ of samples/controls was 24 h. Then, cell fixation began with the removal of the culture medium and the addition of 2% paraformaldehyde (Sigma^®^, St. Louis, MO, USA) for 30 min at 37 °C. Morphology changes were evaluated by: *i*) Hemacolor^®^ rapid staining (Merck^®^, Darmstadt, D.E.) or *ii*) *F*-actin staining with Rhodamine-Phalloidin Reagent (Thermo-Fisher™); according to each manufacturer’s instructions. Later, slides were prepared with Entellan^®^ resin (Merck^®^, Darmstadt, D.E.) or Vectashield^®^/DAPI mounting medium (Vector Laboratories^®^), depending on each assay. Finally, observations were conducted using light microscopy (BX41, Olympus^®^, Miami, FL, USA) or confocal microscopy (LSM 700, Zeiss^®^, Pleasanton, CA, USA) [30].

### 3.9. Experimental Animals

*Nu/Nu* mice were used to determine the antineoplastic activity of RHTR (20 female, 25 ± 5 g weight, and 6-8 weeks old). Mice were maintained in sterile conditions (25 ± 1 °C, and 12 h light/dark cycles) and ad libitum access to food (LabDiet^®^, Richmond, IN, USA) or water in UPEAL (CINVESTAV-IPN). The Institutional Ethics Committee approved all procedures (No. 184-16), following the Official Mexican Regulations: NOM-062-ZOO-1999 [31] and the International Guidelines (see Appendix B, NC3Rs ARRIVE Guidelines Checklist) [32]. Mice were xenotransplanted subcutaneously on the right flank with 5 × 10^6^ SKOV-3 cells. The development of tumor lesions was monitored for 28 days (≈50 mm^3^). Four weeks post-xenotransplantation, rodents were treated daily with 200 mg/kg/i.p. of AE, AF02, or controls for another 28 days. Finally, each week, we evaluated (*i*) bodyweight with an electronic bascule (CS200, Ohaus^®^, Parsippany, NJ, USA) and (*ii*) tumoral lesions with a Vernier caliper (Truper^®^, Jilotepec, Edo. de México, M.X.) [23]. At the end of treatments, mice were anesthetized with 0.1575 mg/250 μL 1× PBS (*v/v*, i.p.) of pentobarbital sodium (PiSA^®^, Guadalajara, Jal., M.X.) for the subsequent analyses (Figure 5). 

### 3.10. Imaging, Paraclinical, and Histopathologic Studies

Computed axial tomography (CAT) was performed with a Somatom Emotion™ (Siemens™) with animals in supine decubitus, full-body, and craniocaudal directions. The sequences and parameters used in the study were modifications based on the ExtrHR protocol. Ultrasonography (USG) was performed in an ultrasound system (LogiQ™ P7, General Electric-Healthcare^®^, Chicago, IL, USA), with real-time image, skeletal muscle preset, and Doppler application. Determination of paraclinical and histopathological studies such as hematic biometry, blood chemistry, and pathological lesions in liver, kidneys, and tumor were performed according to the protocol described by Varela-Rodríguez et al. [23].

### 3.11. Phytochemical Composition of the RHTR Active Fraction

Chromatographic separation was performed by a UPLC-MS system (Acquity™ series-Synapt™ G1, Waters^®^, Milford, MA, USA) in a CSH-C_18_ column (Waters^®^, Milford, MA, USA). Fractions (10 µL, 1 mg/mL) were injected with flow rate of 0.2 mL/min at 30 °C and analyzed using negative/positive-ESI mode (mass range: 50-1500 m/*z*, MS^E^ mode). The mobile phases were water and acetonitrile (solvent B) (MS grade, J.T.Baker^®^, Loughborough, Leicestershire, UK). The separation gradient was as follows: 0 min, 5% B; 0.5 min, 5% B; 20 min, 75% B; 25 min, 75% B; 25.5 min, 90% B. MS data were processed with MassLynx^®^ (version 4.1, Waters^®^, Milford, MA, USA) and Progenesis^®^ QI (version 2.3, Waters^®^, Milford, MA, USA). The principal compounds in the study were confirmed with analytical standards. The procedure was performed following the method described by Varela-Rodríguez [4].

### 3.12. In Silico Studies and Statistical Analysis

The biological activity of the compounds from the RHTR active fraction was predicted with the drug-likeness model of Molsoft© (available from http://molsoft.com/mprop/, accessed on 21 September 2021). The online server uses Lipinski´s criteria (structure–activity relationship, or the rule-of-five) in an algorithm to classify compounds as non-drugs (-X > 0) or drugs (0 < +X) [33,34]. Lipinski´s criteria are used to qualitatively evaluate a compound based on the number of H-bond donors (less than five), H-bond acceptors (less than ten), molecular weight (less than 500 g/mol), or octanol-water partition coefficient (less than five) [35]. For this study, the classification criteria were as follows: a DLMS around 0 to 2 describes a drug-like compound, whereas a DLMS between −6 and −1 corresponds to a non-drug compound [36]. Selected candidates were acquired commercially and evaluated with viability curves in cell culture. Finally, the results are presented as mean ± standard deviation (SD) of three independent assays (*n* = 3; triplicates). The statistical analysis was performed for parametric data with normal distribution using one-way ANOVA and means were compared with normal/pathological controls through the post hoc tests of Tukey–Kramer and Dunnett (Minitab^®^, version 16.1). Differences found were considered as significant when *p* ≤ 0.05.

## 4. Conclusions

The results of the study demonstrate the antineoplastic activity of RHTR in rodents xenotransplanted with ovarian cancer cells mediated by reducing the development of their tumor lesions. Additionally, the selective effect was mainly cytostatic, which agreed with the imaging and histopathological studies. The compounds associated most probably with this therapeutic effect in RHTR are Ga and Myr. Therefore, RHTR, Ga, and Myr could be potentially used as an alternative treatment in ovarian cancer and have a beneficial effect on health; however, more in-depth studies are necessary to understand the molecular mechanism associated with this plant and its compounds.

## 5. Patents

Mexican patent MX/E/2018/078316.

## Figures and Tables

**Figure 1 plants-10-02074-f001:**
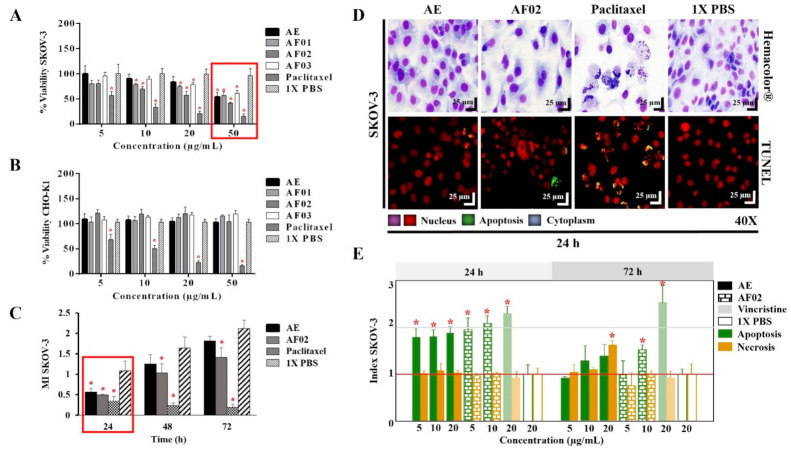
Biological activity of RHTR in ovarian cancer. The IC_50_ of AE and fractions in SKOV-3 (**A**) and CHO-K1 (**B**) cell lines were obtained with dose–response viability curves at 24 h by MTT assay. The antiproliferative (**C**), cytotoxic, and apoptotic (**E**) activities of AE and AF02 were determined in SKOV-3 cells by an ApoTox-Glo™ Triplex Assay. The morphological changes with Hemacolor^®^ rapid staining and TUNEL were observed at 24 h of treatment with AE (50 µg/mL) and AF02 (38 µg/mL), respectively (**D**). The results show the mean ± SD of three biological replicates (*n* = 3, in triplicates). * *p* ≤ 0.05 vs. the control group without treatment (1× PBS) (ANOVA). Paclitaxel or vincristine was used as the positive control. MI, mitotic index (*MI = Abs sample/Abs control*); RHTR, *Rhus trilobata*; AE, aqueous extract; AF, aqueous fraction (numbers indicate the fraction obtained after fractionation on the solid phase with solvents, as described in the Methodology section).

**Figure 2 plants-10-02074-f002:**
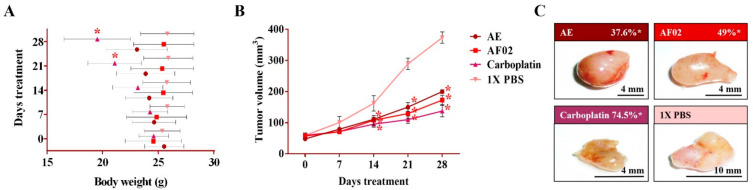
Antineoplastic activity of AE and AF02 from RHTR in ovarian cancer. The bodyweight of rodents was monitored with an electronic bascule for 28 days (**A**). The tumor volume was determined with a Vernier caliper in mice treated with AE and AF02 at 200 mg/kg/i.p./day (Tumoral volume = [Larger diameter × (Shorter diameter)^2^]/2) (**B**). Morphological changes in tumor lesions were analyzed at the end of treatments, and the percent inhibition was calculated (**C**). Results show the mean ± SD of two biological replicates (*n* = 5). * *p* ≤ 0.05 vs. the control group without treatment (1× PBS) (ANOVA). The positive control was carboplatin (50 mg/kg/i.p./3 alternating days per week in mice). RHTR, *Rhus trilobata*; AE, aqueous extract; AF02, aqueous fraction-02 (active fraction obtained with ethyl acetate).

**Figure 3 plants-10-02074-f003:**
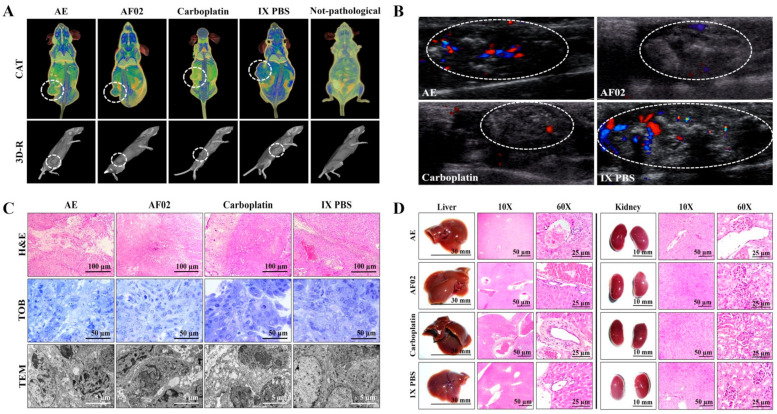
Imaging and histopathologic studies of ovarian tumor lesions and organs of *Nu/Nu* mice treated with RHTR. Analyses with CAT (**A**) and USG (**B**) were performed to observe densitometric and morphological changes in the tumoral lesions, as well as to rule out metastatic processes during the treatments. The tumor lesions are delimited with a white circle. In the Doppler color modality, the arterial and venous flow are indicated in red and blue, respectively. Differential histological patterns were observed in the tumor lesions (**C**) and the absence of tissue damage in the liver and kidneys (**D**) with treatment with AE and AF02 from RHTR at 200 mg/kg/i.p. for 28 days. Results show the mean ± SD of two biological replicates (*n* = 5). The positive control was carboplatin (50 mg/kg/i.p./3 alternating days per week in mice). RHTR, *Rhus trilobata*; AE, aqueous extract; AF02, aqueous fraction-02 (active fraction obtained with ethyl acetate); CAT, computer axial tomographic; 3D-R, 3D reconstruction; USG, ultrasonography; H&E, hematoxylin and eosin; TOB, toluidine blue; MET, transmission electron microscopy.

**Figure 4 plants-10-02074-f004:**
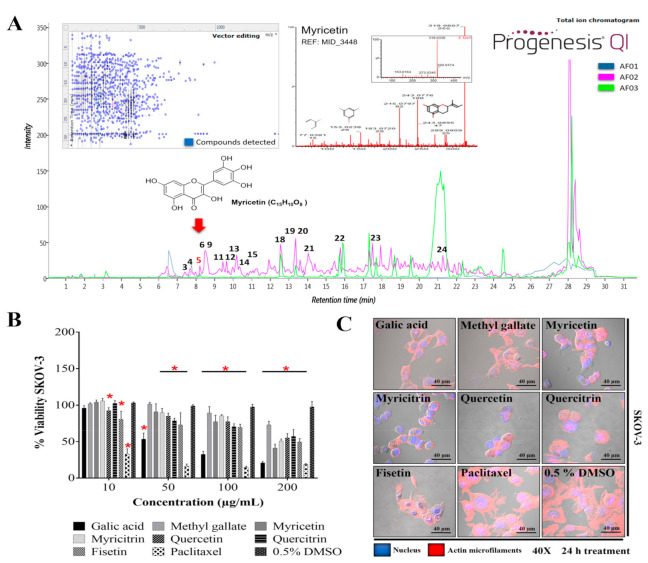
Metabolite profile of RHTR fractions and biological activity in cell lines. Total ion chromatograms (ESI+) for AF01, AF02, and AF03 revealed the presence of myricetin (RT: 8.27 min) in all of them (**A**). Compounds with higher relative abundance in AF02 are indicated (**A**). The vector map indicates 215 putative identifications of 555 compounds found in AF02 (**A**). Comparisons with the fragmentation pattern of myricetin in AF02 and an analytical standard (**A**). The IC_50_ of metabolites from RHTR was determined by dose–response viability curves with an MTT assay for 24 h in SKOV-3 cells (**B**), and morphological changes were determined by immunofluorescence with phalloidin-rhodamine (actin microfilaments) and DAPI (nucleus) in the same conditions (**C**). Results show the mean ± SD of three biological replicates (*n* = 3, in triplicates); * *p* ≤ 0.05 vs. the vehicle group without treatment (0.5% DMSO) (ANOVA). RHTR, *Rhus trilobata*; AF, aqueous fraction (numbers indicate the fraction obtained after fractionation on solid-phase with solvents, as described in the Methodology).

**Figure 5 plants-10-02074-f005:**
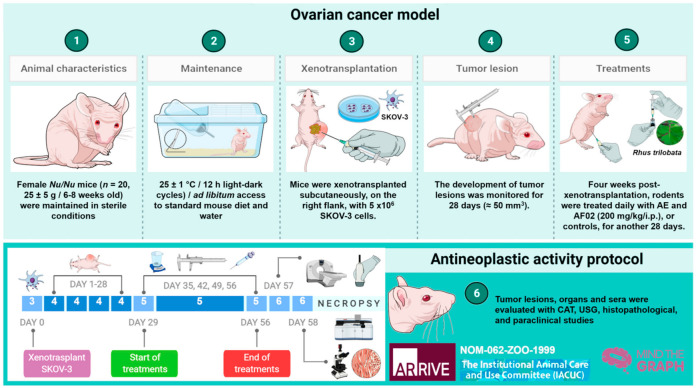
Timeline of the antineoplastic protocol implemented for RHTR treatments. The antineoplastic activity of AE and FA02 was determined in female *Nu/Nu* mice xenotransplanted with SKOV-3 cells, according to the Official Mexican Regulations [31] and International Guidelines [32], as well as the methodology proposed previously by Varela-Rodríguez et al. (2020) [23]. RHTR, *Rhus trilobata*; AE, aqueous extract; AF02, aqueous fraction-02 (active fraction obtained with ethyl acetate).

**Table 1 plants-10-02074-t001:** Morphological characteristics of ovarian tumor lesions treated with RHTR.

Treatments	AE	AF02	Carboplatin	1× PBS
Weight (g)	0.24 ± 0.09 *	0.19 ± 0.10 *	0.14 ± 0.06 *	0.34 ± 0.09
Larger diameter (mm)	9.07 ± 0.25 *	7.40 ± 0.95 *	3.70 ± 0.26 *	14.53 ± 2.4
Tumor volume (mm^3^)	201.4 ± 11.6 *	172.7 ± 15.3 *	137.3 ± 18.6 *	373.3 ± 18.1
Attenuation coefficient (HU)	47 ± 2.16	48.8 ± 0.83	48.7 ± 0.95	47 ± 2.1
Vascularity	+	+	+	+++
Fibrosis	+	+	+	+
Morphology	Cystic	Cystic	Solid	Cystic

Results show the mean ± SD of two biological replicates (*n* = 5). * *p* ≤ 0.05 vs. control values treated with 1× PBS vehicle (ANOVA). The positive control was carboplatin (50 mg/kg/i.p./3 alternating days per week in mice). HU, Hounsfield unit; +, present; +++, abundant. RHTR, *Rhus trilobata*; AE, aqueous extract; AF02, aqueous fraction-02 (active fraction).

**Table 2 plants-10-02074-t002:** Paraclinical studies in *Nu/Nu* mice treated with RHTR.

Treatments	AE	AF02	Carboplatin	1× PBS	Reference Range (mean)
Glucose (mg/dL)	72 ± 23.9	114 ± 21.2 *	107 ± 14.8 *	78.3 ± 16.6	63–176 (89)
Triglycerides (mg/dL)	71.5 ± 26.2	80 ± 11.35	103.7 ± 12.7 *	87.6 ± 12.01	55–115 (85)
Cholesterol (mg/dL)	66 ± 25.4 *	62 ± 2.52 *	108.6 ± 19.4 *	85.5 ± 13.4	26–82 (64)
Protein (g/dL)	4.48 ± 0.6	4.42 ± 0.62	5.1 ± 1.46	4.7 ± 0.56	4–8.6 (6.2)
Albumin (g/dL)	2.1 ± 0.8 *	2.03 ± 0.35 *	1.33 ± 0.29	1.36 ± 0.21	2.5–4 (3.2)
AST (GOT) (U/L)	132.6 ± 45.7	145.7 ± 7.69	162.9 ± 8.2	149.8 ± 27.29	55–251 (139)
ALT (GPT) (U/L)	41.2 ± 9.8	32.43 ± 19.2	64.4 ± 6.87 *	40.75 ± 14.5	17–77 (47)
Total bilirubin (mg/dL)	0.47 ± 0.07 *	0.39 ± 0.08 *	0.58 ± 0.2	0.67 ± 0.16	0.20–1.0 (0.6)
Alkaline phosphatase (U/L)	46.5 ± 23.2 *	83.6 ± 24.04	66.6 ± 15.6	71 ± 10.36	9–88 (48.5)
Creatinine (mg/dL)	0.74 ± 0.4	0.7 ± 0.25	0.37 ± 0.26 *	0.65 ± 0.2	0.2–0.9 (0.5)
Urea (mg/dL)	40.8 ± 8.2	57.6 ± 10.7	59.4 ± 9.99	47.9 ± 7.16	46.9–73 (60.1)
BUN (g/dL)	19 ± 3.8	26.8 ± 4.7	27.8 ± 4.63	22.36 ± 3.32	11–27 (19)
Hemoglobin (g/dL)	15.2 ± 0.98	12.35 ± 2.2	13.1 ± 1.61	14.4 ± 2.13	10–17 (13.1)
Hematocrit (%)	48.3 ± 7.92	43.6 ± 17.8	40.57 ± 4.16	44.52 ± 2.17	39–49 (40.4)
Erythrocytes (×10^6^/mm^3^)	7.24 ± 1.06	7.67 ± 2.26	7.81 ± 0.88	7.92 ± 1.93	8.3
Leukocytes (×100/mm^3^)	3600 ± 903 *	4242.6 ± 980	2750 ± 777 *	5720 ± 684	5–12 (6.33)
Platelets (×10^6^/μL)	736 ± 1.414	689 ± 5.425	759 ± 19.143	710 ± 6.716	116
PCT	0.53 ± 0.07	0.49 ± 0.10	0.55 ± 0.29	0.55 ± 0.30	--

Results show the mean ± SD of two biological replicates (*n* = 5). *, *p* ≤ 0.05 vs. control values treated with 1× PBS vehicle (ANOVA). The positive control was carboplatin (50 mg/kg/i.p./3 alternating days per week in mice). Reference range provided by the paraclinical laboratory (Merasoma Laboratory), minimum and maximum normal value for the analyte of interest in mice, and the respective midrange [11,12]. RHTR, *Rhus trilobata*; AE, aqueous extract; AF02, aqueous fraction-02 (active fraction); AST (GOT), aspartate aminotransferase; ALT (GPT), alanine aminotransferase; BUN: blood urea nitrogen; PCT: plateletcrit.

**Table 3 plants-10-02074-t003:** Major phytochemical compounds in AF02-RHTR by UPLC-MS^E^.

No.	RT (min)	PI	Compound Name	Biological Activity	Presence in Plants	DLMS
1	6.18	HMDB41635	myricetin 3-arabinoside	Antioxidant, anticarcinogenic	*Rhus* spp.	0.90
2	6.44	CSID70398	methyl gallate *	Antioxidant	*Rhus* spp.	−0.49
3	7.05	HMDB38831	epigallocatechin 3-cinnamate	Antioxidant, antibacterial	*Ocotea porosa*	0.24
4	7.36	CSID3525015	3,5-digalloylepicatechin	Antioxidant	--	1.52
5	8.27	CHEBI:18152	myricetin *	Antioxidant, antineoplastic	*Rhus* spp.	−0.04
6	8.47	LMPK12112436	myricitrin *	Antioxidant, antineoplastic	*Rhus* spp.	0.78
7	8.52	LMPK12110568	quercetin 3-(2’’’-galloylglucosyl)-(1→2)-alpha-L-arabinofuranoside	Antioxidant	*Euphorbia pachyrhiza*	1.24
8	8.52	CHEBI18082	1,2,3,4,6-pentakis-*O*-galloyl-*β*-D-glucose	Antineoplastic	*Rhus* spp.	0.35
9	8.60	CHEBI:16243	quercetin *	Antioxidant, anti-inflammatory, antineoplastic	*Rhus* spp.	0.93
10	9.13	CSID24784962	4-*O*-digalloyl-1,2,3,6-tetra-*O*-*β*-D-galloylglucose	Antibacterial, antineoplastic, antithrombotic	*Rhus typhina*	0.35
11	9.45	CHEBI:5715	hibiscoquinone a	Antioxidant	*Hibiscus* spp.	−0.52
12	9.45	LMPK12112171	quercitrin *	Antioxidant, antineoplastic	*Rhus* spp.	1.04
13	9.96	LMPK12112447	myricetin 3-(4’’-galloylrhamnoside)	Antioxidant	--	1.12
14	10.20	LMPK12100067	obtusaquinol	Antiparasitic, nitric oxide inhibitor	*Dalbergia* spp.	−0.03
15	10.94	CSID35013429	epifisetinidol-(4*β*→8)-catechin	Antibacterial, *α*-amylase and lipase inhibitor	*Cotinus coggyria*	0.61
16	11.92	LMPK12111566	fisetin *	Antineoplastic, antioxidant, antiangiogenic	*Rhus* spp.	0.76
17	12.54	LMFA01010048	margaric acid	Antioxidant, antifungal	*Rhus typhina*	−0.33
18	12.54	LMFA01050146	12*S*-hydroxy-16-heptadecynoic acid	Anti-inflammatory	--	−0.38
19	13.37	CSID8034429	(-)-pinellic acid	Metabolism, adjuvant activity	*Pinelliae tuber*	−1.08
20	13.37	LMFA01030130	11,14-eicosadienoic acid	Antioxidant, anti-inflammatory	*Rhus typhina*	−0.08
21	13.9	LMPK12040009	amentoflavone	Antioxidant, anti-inflammatory	*Rhus* spp.	0.51
22	15.94	CSID10297786	3-(1,1-dimethylallyl)-8-(3,3-dimethylallyl) xanthyletin	--	*Ruta graveolens*	−1.31
23	17.7	CHEBI:31014	lignocerate	Anti-inflammatory	*Oleandra neriiformis*	−0.33
24	21.28	LMFA02000057	2*R*-hydroxy-9*Z*,12*Z*-octadecadienoic acid	Anti-allergic	*Brassica campestris*	−0.96

Results of three biological replicates (*n* = 3, in triplicates). Compounds putatively identified based on the fragmentation pattern and metabolic databases (AraCyc, PlantCyc, KEGG). DLMS: –6 to –1, non-drug compound / 0 to 2, drug-like compound. * Compounds confirmed with analytical standards. RHTR, *Rhus trilobata*; AF02, aqueous fraction-02 (active fraction); RT, retention time (min); PI, putative identity; DLMS, drug-likeness model score.

**Table 4 plants-10-02074-t004:** Biological and cytotoxic activity of metabolites from RHTR in cell lines.

Sample/Cell Line	SKOV-3	OVCAR-3	CACO-2	BEAS-2B	CHO-K1
Gallic acid	50 (294)	43 (253)	25 (147)	25 (147)	100 (588)
Methyl gallate	>200 (1,086)	>200 (1,086)	100 (543)	200 (1,086)	130 (706)
Myricetin	166 (522)	94 (295)	62 (195)	64 (201)	33 (104)
Myricitrin	197 (424)	200 (431)	>200 (431)	160 (344)	94 (202)
Quercetin	>200 (662)	200 (662)	150 (496)	189 (625)	127 (420)
Quercitrin	>200 (446)	>200 (446)	179 (399)	>200 (446)	>200 (446)
Fisetin	200 (699)	200 (699)	100 (349)	50 (175)	46 (161)
Paclitaxel	7 (8.20)	8 (9.37)	20 (23.42)	8 (9.37)	10 (11.71)

Results show the half-maximal inhibitory concentration (IC_50_, μg/mL and [µM]) obtained of three biological replicates (*n* = 3, in triplicates). The positive control was paclitaxel.

## Data Availability

All data generated or analyzed during this study are included in this published article (as well as Appendix A). Raw data are available from the corresponding author on reasonable request.

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
