# Peer review of "Antineoplastic Activity of Rhus trilobata Nutt. (Anacardiaceae) against Ovarian Cancer and Identification of Active Metabolites in This Pathology"

_plants, 2021, doi:10.3390/plants10102074_

Round 1
Reviewer 1 Report
Dear authors,
my main comments is related to the design of the study and the bias you have made which happens often when it comes to the work with mixtures of natural products and their biological activity. As you have said in line 235/236:
"In silico analysis and bibliographic search of the medici- 235 nal properties of major metabolites in AF02-RHTR were carried out to determine its pos- 236 sible activity against ovarian cancer"
I would love to comment I am well aware of the fact that many studies related to the natural compounds and their biological activity focuses on the most abundant natural products instead of watching the whole picture and respecting the pharmacological principle where even some molecule has a small amount in plant extract, it can have a powerful effect. Not to mention the synergism where there are a number of studies showed that the metabolites in the mixture have a more powerful effect, than isolated and tested as pure. Therefore focusing only on the most abundant metabolites and studying them in more detail, creates a bias and potentially wrong conclusions if you ignore the rest of the found metabolites, and you have analyzed 24 compounds. Could you please write something like a disclaimer where you state that you are aware of the fact there might be some more metabolites responsible for the activity, or synergism, whilst you have a focus on these, but there is still space for more in-depth studies? I know you have written something like this in relation to the selected pure compounds, but still I would love you to clear out that the design of the study may have created some bias and there might be some more metabolites worth studying. Besides this, I don't have much of the comments since the writing was clear, methods well described, results well presented.
Author Response
Answer:
We would like to thank Reviewer 1 for his/her comments and observations to improve this article. We hope to accommodate your requests.
For this purpose, the following modifications were made to the manuscript:
The authors have added a complete paragraph in the manuscript to mention the following:
Page 7, lines 233-234
“To delimit the scope of this study, the experimental design was focused on an …”
Page 7, lines 236-240
“However, the choice of this design most likely generated a bias in which possibly, unintentionally, some active compound in low concentration or the synergistic effect that can be generated by the combined presence of several compounds were omitted. This opens the possibility for future studies aimed at elucidating these two possibilities, either by our own group or by members of the community interested in studying natural compounds with potential anticancer bioactivity.
We also would like to point out that this study is the first approach to understanding the antineoplastic activity of RHTR. Therefore, the selection and implementation of the experimental design mentioned in the manuscript intended to simplify obtaining results for this purpose and thus elucidate the pharmacological potential of this plant, avoiding the complexity of its nature. We thus thank Reviewer 1 for his/her valuable comments. We hope to be addressing the medicinal properties of RHTR in a more general way soon.
Reviewer 2 Report
After reviewing the article entitled: “Antineoplastic activity of Rhus trilobata Nutt. (Anacardiaceae) against ovarian cancer and identification of active metabolites 3 in this pathology”. I noted a high quality of the work and well written and planned.
The research is focused on Rhus trilobata (RHTR), which is a medicinal plant with cytotoxic activity in different cancer cell lines. However, the active compounds in this plant against ovarian cancer are unknown. This study aimed to evaluate the antineoplastic activity of RHTR and identify its active metabolites against ovarian cancer. As results: The metabolite profile of AF02 showed a higher abundance of flavonoid and lipid compounds compared with AE by UPLC-MSE. Gallic acid and myricetin were the most active compounds in RHTR against SKOV-3 cells with 50/166 μg/mL (p ≤ 0.05, ANOVA). Antineoplastic studies in Nu/Nu female mice with subcutaneous SKOV-3 cells xenotransplant revealed that 200 mg/kg/i.p. of AE and AF02 inhibited ovarian tumor lesions from 37.6 to 49 % after 28 days (p ≤ 0.05, ANOVA). In conclusion, RHTR has antineoplastic activity against ovarian cancer by a cytostatic effect related to gallic acid and myricetin. Therefore, RHTR could be a complementary treatment for this pathology.
Abstract, introduction, and results as well as discussion are well organized and mainly results support the scientific evidence about this species with potential antineoplastic activity in preclinical phase.
However, I have minor comments:
- About the in-silico study, which in your methodology (pag 14 line 430) is stated that some bioactive molecules were evaluated and there is a scale to consider it as toxic or non-toxic. Please, include a reference to support this information.
- In table 2: What was the reference to include range reference in mice? Or, they were proportioned by the Bioterio, if it was the second, please, include it as a caption.
Author Response
Reviewer 2 comments:
- About the in-silico study, which in your methodology (pag. 14 line 430) is stated that some bioactive molecules were evaluated and there is a scale to consider it as toxic or non-toxic. Please, include a reference to support this information.
- In table 2: What was the reference to include range reference in mice? Or, they were proportioned by the Bioterio, if it was the second, please, include it as a caption.
Answers:
We would like to thank Reviewer 2 for his/her comments and observations to improve this article. We hope to accommodate your requests.
For this purpose, the following modifications were made to the manuscript:
- About the in-silico study, which in your methodology (pag 14 line 430) is stated that some bioactive molecules were evaluated and there is a scale to consider it as toxic or non-toxic. Please, include a reference to support this information.
We regret the confusion. In this sense, the methodological section: 3.12. In silico studies and statistical analysis, was modified to be more precise about the methodological basis and selection criteria of the compounds analyzed in the study. Whereby, the following paragraph was added:
Page 13 and 14, lines 430-438
“The online server uses Lipinski´s criteria (structure–activity relationship, or the rule-of-five) in an algorithm to classify compounds as non-drugs (-X > 0) or drugs (0 < +X) [33, 34]. Lipinski´s criteria are used to qualitatively evaluate a compound based on the number of H-bond donors (less than five), H-bond acceptors (less than ten), molecular weight (less than 500 g/mol), or octanol-water partition coefficient (less than five) [35]. For this study, the classification criteria were as follows: a DLMS around 0 to 2 describes a drug-like compound, whereas a DLMS between −6 and −1 corresponds to a non-drug compound [36]”.
Likewise, the corresponding references were added in the manuscript
Page 16, lines 576-584
References:
- Molsoft LLC. © (2021). Drug-Likeness and molecular property prediction. http://molsoft.com/mprop/ (Accessed: 21 Sep 2021).
- Hussein W., SaÄŸlık B.N., Levent S., Korkut B., Ilgın S., Özkay Y. & Kaplancıklı Z.A. (2018). Synthesis and biological evaluation of new cholinesterase inhibitors for Alzheimer's disease. Molecules, 23(8): 2033. DOI: 10.3390/molecules23082033.
- Lipinski C.A., Lombardo F., Dominy B.W. & Feeney P.J. (2001). Experimental and computational approaches to estimate solubility and permeability in drug discovery and development settings. Advanced Drug Delivery Reviews, 46(1-3): 3–26. DOI: 10.1016/s0169-409x(00)00129-0
- Macalalad M. & Gonzales-3rd A.A. (2021). In-silico screening and identification of phytochemicals from Centella asiatica as potential inhibitors of sodium-glucose co-transporter 2 for treating diabetes. Journal of Biomolecular Structure & Dynamics, 2021: 1–18. DOI: 10.1080/07391102.2021.1969282.
And the Table 3-footer was modified according to the reviewer comment:
Page 8, line 252
“DLMS: -6 to -1, non-drug compound/0 to 2, drug-like compound".
- In table 2: What was the reference to include range reference in mice? Or, they were proportioned by the Bioterio, if it was the second, please, include it as a caption.
In the manuscript, Table 2-footer was modified to add the following sentence:
Page 6, line 203
“Reference range provided by the paraclinical laboratory (Merasoma Laboratory) …”